# Bowel Dilatation on Initial Plane Abdominal Radiography May Help to Assess the Severity of Necrotizing Enterocolitis in Preterm Infants

**DOI:** 10.3390/children7020009

**Published:** 2020-01-23

**Authors:** Zlatan Zvizdic, Irmina Sefic Pasic, Amra Dzananovic, Nedzad Rustempasic, Emir Milisic, Asmir Jonuzi, Semir Vranic

**Affiliations:** 1Clinic of Pediatric Surgery, University Clinical Center Sarajevo, 71000 Sarajevo, Bosnia and Herzegovina; zlatan.zvizdic@gmail.com (Z.Z.); emirilejla@gmail.com (E.M.); jonuziasmir@hotmail.com (A.J.); 2Department of Radiology, University Clinical Center Sarajevo, 71000 Sarajevo, Bosnia and Herzegovina; irmina.sefic@gmail.com (I.S.P.); amradzananovic@gmail.com (A.D.); 3Clinic of Cardiovascular Surgery, University Clinical Center Sarajevo, 71000 Sarajevo, Bosnia and Herzegovina; nrustempasic@yahoo.com; 4College of Medicine, QU Health, Qatar University, Doha PO Box 2713, Qatar

**Keywords:** necrotizing enterocolitis, abdominal radiography, bowel distension, disease severity

## Abstract

Background: Necrotizing enterocolitis (NEC) is the most common life-threatening gastrointestinal emergency associated with prematurity. Timely diagnosis and adequate treatment are crucial to reduce the morbidity and mortality of the affected infants. The aim of this study was to evaluate the diagnostic yield of bowel dilatation on plane abdominal radiography (AR) in the early diagnosis and NEC severity in preterm infants. Methods: We retrospectively reviewed initial ARs of 50 preterm infants with NEC ≥ stage II admitted to the neonatal intensive care unit (NICU) in a tertiary-care hospital. The largest bowel loops diameters (AD), the latero-lateral diameters of the peduncle of the first lumbar vertebra (L1), and the distance of the upper edge of the first lumbar vertebra and the lower edge of the second one, including the disc space (L1–L2), were measured. All anteroposterior ARs were done in a supine projection on the day of onset of the initial symptoms of NEC. Results: Preterm infants with surgical NEC showed a statistically significant increase in the AD/L1 ratio (*p* < 0.001) and AD/L1-L2 ratio (*p* < 0.001) compared with preterm infants with medical NEC. We found no significant association between the site of the most distended bowel loop and the severity of NEC (*p* > 0.05). Conclusion: Bowel loop distension on initial AR may serve as an additional diagnostic tool in the early diagnosis and severity of stages II/III NEC. Further prospective clinical studies should validate the results from this study.

## 1. Introduction

Necrotizing enterocolitis (NEC) is the most common life-threatening gastrointestinal emergency in newborns that may require surgical treatment [1]. The disease predominantly affects premature infants, particularly those with a low birth weight and lower gestational age. The mortality rate for NEC with perforation is approximately 30% [2]. Delay of NEC diagnosis and treatment may aggravate this acquired medical condition. Aggravating circumstances in the timely diagnosis of NEC are that it is highly variable, nonspecific and has subtle initial signs and symptoms, and the lack of single imaging test that is sensitive and specific enough for the diagnosis of NEC [3].

NEC diagnosis is based on the evidence of systemic and intestinal signs. Systemic signs include temperature instability, apnea, bradycardia, episodes of oxygen desaturation, and signs of lethargy or irritability, while intestinal signs include increased pregavage residuals, bilious aspirates, abdominal distention, emesis, guaiac-positive stool, absent bowel sounds, abdominal tenderness, and a right lower quadrant mass [1]. However, the clinical presentation of NEC may substantially vary and may include some, all, or none of the above-mentioned systemic or intestinal signs [1].

Because of its widespread availability and low cost, abdominal radiographs (ARs) are commonly used in the initial evaluation of patients with suspected NEC, and later on, in assessment of the progression of the disease. Although nonspecific, AR findings of intestinal distension are frequently one of the earliest signs of NEC, but it is occasionally difficult to assess the extent of intestinal distension.

The aim of this study was to evaluate the clinical utility of bowel dilatation on plane AR in the early diagnosis and assessment of NEC severity in preterm infants.

## 2. Materials and Methods 

The study included analysis of ARs from 50 preterm infants with NEC: 29 with medical and 21 with surgical NEC (≥stage II). All the patients were treated at the neonatal intensive care unit in the period 2008–2012 (please see the flowchart below).

The clinical characteristics of the cohort were previously reported [4]. The study was conducted in accordance with the Declaration of Helsinki 1964 and the local institutional review board approved the study (Approval number: 0305-11118/2010) [4].

NEC patients were staged according to the criteria that were previously proposed by Bell and modified by Walsh [5]. Stage II NEC (medical NEC, Sarajevo, Bosnia and Herzegovina) was defined according to the onset of at least one of the below clinical symptoms: Abdominal distension, emesis, and occult or gross blood in the stool, and radiographic or ultrasound findings of pneumatosis intestinalis or portal vein gas. Stage III NEC (surgical NEC, Sarajevo, Bosnia and Herzegovina) was defined according to the above clinical symptoms plus radiological (radiographic or ultrasound) findings of pneumoperitoneum or a gasless abdomen or someone who requires surgery if medical therapy was not valid within 48 h [5]. Those excluded were all preterm infants who did not have results for an initial abdominal radiography (*n* = 2) and preterm infants showing pneumoperitoneum on initial abdominal radiography (*n* = 3) (Figure 1).

The radiological assessment included ARs (supine anteroposterior projection) obtained as a first radiology method. Only the first abdominal radiographs taken for a patient with suspected necrotizing enterocolitis was re-reviewed for the study purposes. We examined the ratio between the diameter of the most distended intestinal loop (AD) and the diameter of the lateral edges of the first lumbar vertebral body (L1) as previously suggested by Edwards and Martins [6,7] (Figure 2). In addition, we explored the ratio between the upper edge of the first lumbar vertebra and the lower edge of the second one, including the disc space (L1–L2) (Figure 2).

AD—Diameter of the most distended intestinal loop (14.7 mm).L1—Diameter of the lateral edges of the first lumbar vertebral body (17.8 mm).L1–L2—The ratio between the upper edge of the first lumbar vertebra and the lower edge of the second one, including the disc space (17.3 mm).

Two experienced pediatric radiologists and surgeons respectively performed and analyzed the millimeter ruler measurements. The results of the measurements on ARs, designed and presented to minimize possible differences in height, weight, and age of the preterm infants included in the study, were compared between preterm infants with different stages of NEC (medical versus surgical NEC). The anatomical regions where the most distended bowel loops were found were classified as the right hypochondrium, left hypochondrium, right flank, left flank, right iliac fossa, and left iliac fossa. AR was performed in a supine position using a single X-ray unit, GE TMX+ (General Electric, Boston, MA, USA), and Agfa CR30-X computed radiography (CR) imaging system (Agfa-Gevaert, Mortsel, Belgium). The exposure parameters used were: AP projection with a tube potential (kV) of 53 and tube loading (mAs) of 3.2; and a focus–skin distance (cm) of 87.7 and CR detector size (cm^2^) of 18 × 24. All the measurements were done independently and in case of discrepancies, the researchers would discuss it to reach a consensus.

For statistical analysis, we used the non-parametric Mann–Whitney U test to compare the differences in the examined variables. All the tests were done using Statistical Package for the Social Sciences (SPSS) IBM Version 26 (UNICOM Systems, Inc., Mission Hills, CA, USA). *p*-values < 0.05 were considered significant.

## 3. Results

The descriptive data related to the AR measurements and ratios are summarized in Table 1. Figure 1 shows an example of the measurements that were done on the abdominal radiograph.

Preterm infants with surgical NEC showed a statistically significant increase in the AD/L1 ratio compared with the infants with medical NEC (*p* < 0.001) (Table 1). Similarly, the AD/L1–L2 ratio was significantly higher in surgical compared with patients having medical NEC (*p* < 0.001) (Table 1).

Although the most distended bowel loops were located in the right iliac fossa (29.4%) and in the left flank (24%) in >50% of preterm infants with NEC, no significant association was found between the site of the most distended bowel loop and the severity of NEC (*p* > 0.05).

## 4. Discussion

The plane AR has traditionally been used as the main imaging modality in the assessment of NEC [8]. Clinical presentation along with the radiological findings of intestinal pneumatosis, hepatic portal venous gas, and pneumoperitoneum are highly suggestive of NEC. However, significant variations in the interpretation of AR in NEC diagnostics have also been widely recognized; hence, ARs alone are not sufficient to make a correct and timely diagnosis of NEC [9]. Keeping this in mind and to minimize the interobserver variability in the interpretation of abdominal radiographics, two experienced pediatric radiologists and surgeons respectively interpreted all ARs, first individually and then by consensus.

On initial AR, focal or diffuse gaseous distension of the intestine may indicate early or developing NEC [9]. However, intestinal distension may also be seen in preterm infants without NEC, particularly in the first two weeks of life [9]. Preterm infants on ventilation, especially on nasal continuous positive airway pressure, may also have similar AR presentation [10]. AR tends to have a high positive predictive value but low sensitivity (<50%) for radiographic signs of NEC [11]. Thus, Tam et al. reported that AR had a sensitivity to diagnose pneumatosis in 44%, portal venous gas in 13%, and free air in 52% of cases [11]. The authors pointed out that the negative radiological findings need to be cautiously interpreted in preterm infants with suspected NEC [11].

Despite all AR limitations related to NEC, there has been an intention for decades to redefine the role of AR and use it as an additional tool in diagnosing NEC and assessing the severity of the disease. In this regard, Edwards suggested that the measured diameter of bowel loops should be compared to relatively fixed bone structures, such as the upper lumbar spine [6]. This anatomical area was selected because of its visibility on all ARs, and because the length of the lumbar spine closely correlates with the fetal size. In addition, the configuration of the bones would not be expected to change appreciably with respiration [12]. Edwards also suggested that the previously used subjective description of bowel distension (mild/moderate/severe) should be replaced by numerical values [6]. A similar proposal came from Martins et al., who concluded that measurement of the most dilated bowel loop on initial supine ARs is a simple and reproducible method that adds diagnostic and prognostic information [7]. The authors also showed that bowel dilatation was associated with a worse prognosis in NEC patients [7]. Using these two approaches, we provide further evidence of a positive relationship between the diameter of the most distended bowel loop with the transverse diameter of the first lumbar vertebra and the diameter of the distance of the first two lumbar vertebrae in the prediction of the occurrence of NEC and its severity. However, our results regarding the association of the site of the most distended bowel loop and the severity of the disease did not reach statistical significance, which is in line with the study of Martins et al. [7]. The reason for the right iliac fossa as the most common localization (~30%) of the distended bowel loop could be due to the difficulties of collateral circulation in the terminal ileum region [7]. Our study has several limitations. Because of the retrospective nature of the study, the interpreters were not fully blinded to the patients’ outcome, which may represent a potential bias. We also focused on stages II and III of NEC and included a relatively small number of patients.

## 5. Conclusions

We conclude that our data further support the clinical utility of AR and the proposed diameters/ratios in assessing the severity of stages II/III NEC. However, larger and prospective clinical studies are required to confirm our results and their clinical utility.

## Figures and Tables

**Figure 1 children-07-00009-f001:**
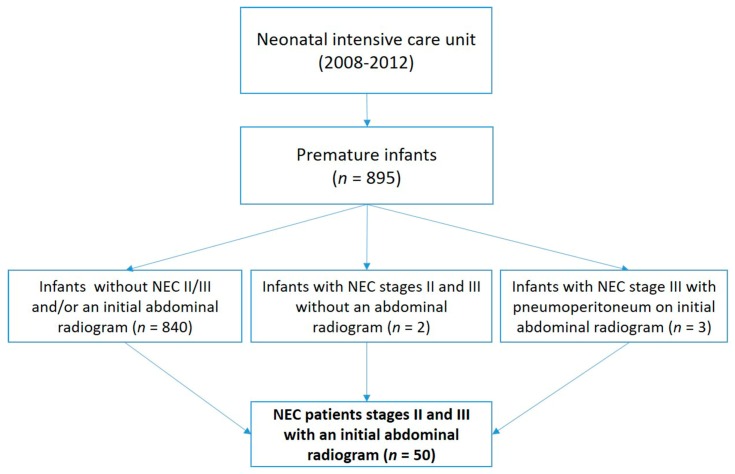
A flowchart that shows the selection of the necrotizing enterocolitis (NEC) patients for the current study.

**Figure 2 children-07-00009-f002:**
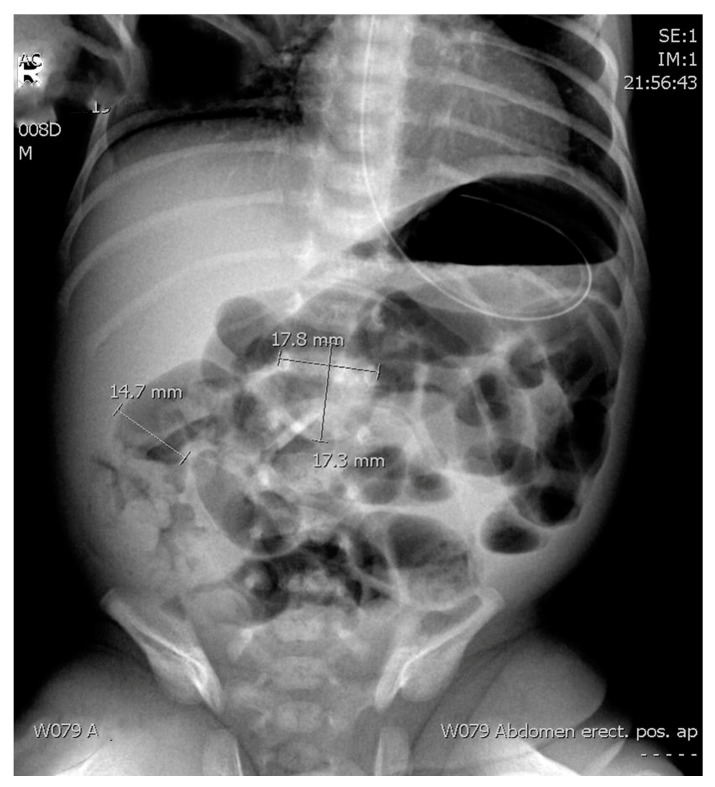
Supine anteroposterior projection abdominal radiography of a preterm infant with necrotizing enterocolitis with the measurement mode shown.

**Table 1 children-07-00009-t001:** The comparisons of the assessed parameters on the abdominal radiographs in the preterm infants with medical and surgical NEC.

**The Ratio 1 ***		
	**Medical NEC (*n* = 29) AP/L1**	**Surgical NEC (*n* = 21) AP/L1**
Minimum	0.740	0.790
Maximum	1.000	1.100
Mean	0.864	0.992
95% CI	0.834–0.893	0.962–1.022
**The Ratio 2 ****		
Minimum	0.820	0.830
Maximum	1.180	1.220
Mean	0.944	1.112
95% CI	0.910–0.978	1.080–1.165

* The ratio 1 pertains to the ratio between the diameter of the most distended intestinal loop (AD) and the diameter of the lateral edges of the first lumbar vertebral body (L1). ** The ratio 2 pertains to the ratio between the upper edge of the first lumbar vertebra and the lower edge of the second one, including disc space (L1–L2). *n* = number; CI = confidence interval; AP = Anteroposterior; L = lumbar. NEC = necrotizing enterocolitis.

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
