# Peer review of "Bowel Dilatation on Initial Plane Abdominal Radiography May Help to Assess the Severity of Necrotizing Enterocolitis in Preterm Infants"

_children, 2020, doi:10.3390/children7020009_

Round 1
Reviewer 1 Report
Authors have aimed to study the potential of the ratio of bowel dilatation vs lumbar vertebra as a tool for early diagnosis of NEC and to assess disease severity. This is a relevant topic, since timely diagnosis of NEC is associated with a better prognosis. I have some major and minor revisions for authors before this article can be considered for publication in Children.
Major revisions:
In the Material and Methods section, authors make no mention of the timing of the abdominal radiography, when multiple radiographic images were made, how did authors select the radiographic image of interest?
How were medical and surgical NEC defined, what kind of infants were included in both groups? What happened to the infants that were too sick to undergo surgical treatment? Authors should elaborate more on this.
Lines 65-6: I am not sure what authors are trying to outline here, please clarify.
Lines 106-107: Here, authors make mention of the variation in interpretation of the AR, however, authors make no mention of there was overall consensus between the radiologists and surgeons in the measurements of the dilatations. Authors should elaborate more on this.
Lines 107-108: Here, authors mention that two experienced pediatric radiologists and surgeons interpreted all ARs, however, was this to limit the influence of having only one observer? How did authors handle interobserver variation? Please clarify
Minor revisions
Table 1. only those numbers that were used in the analyses should be displayed, I am not sure why authors choose to report both median and mean. I would advise authors to keep the table simple ad include only those results/numbers that were relevant for the analysis.
Lines 123-124: “a” should be removed from the sentence
Lines 127-128: “worst” should be replaced by “worse”.
Author Response
Authors have aimed to study the potential of the ratio of bowel dilatation vs lumbar vertebra as a tool for early diagnosis of NEC and to assess disease severity. This is a relevant topic, since timely diagnosis of NEC is associated with a better prognosis. I have some major and minor revisions for authors before this article can be considered for publication in Children.
Major revisions:
In the Material and Methods section, authors make no mention of the timing of the abdominal radiography, when multiple radiographic images were made, how did authors select the radiographic image of interest?ANSWER: Only the first abdominal radiograph taken for a patient with suspected NEC was re-reviewed and used for the current study. For clarity reasons, we added a flowchart (Figure 1) to show our approach and selection of the patients for the present study.
How were medical and surgical NEC defined, what kind of infants were included in both groups? What happened to the infants that were too sick to undergo surgical treatment? Authors should elaborate more on this.ANSWER: Absolutely agree and sorry for not adding it in the first draft. We now incorporated the definitions of both medical and surgical NEC (please see page 2-3, lines 64-9).
Lines 65-6: I am not sure what authors are trying to outline here, please clarify.ANSWER: We removed this paragraph.
Lines 106-107: Here, authors make mention of the variation in interpretation of the AR, however, authors make no mention of there was overall consensus between the radiologists and surgeons in the measurements of the dilatations. Authors should elaborate more on this.ANSWER: We now added a paragraph on this important issue in both materials and methods (lines 96-8) and discussion (lines 129-131).
Lines 107-108: Here, authors mention that two experienced pediatric radiologists and surgeons interpreted all ARs, however, was this to limit the influence of having only one observer? How did authors handle interobserver variation? Please clarifyANSWER: Thanks for this important note. We explored this in more detail in Materials and Methods (lines 96-8) and briefly discussed (lines 129-131).
Minor revisions
Table 1. only those numbers that were used in the analyses should be displayed, I am not sure why authors choose to report both median and mean. I would advise authors to keep the table simple ad include only those results/numbers that were relevant for the analysis.ANSWER: Thanks for this comment. The redundant descriptive data are no removed and table 1 has been modified accordingly.
Lines 123-124: “a” should be removed from the sentence
ANSWER: Done.
Lines 127-128: “worst” should be replaced by “worse”.
ANSWER: Done.
Reviewer 2 Report
I’m somewhat puzzled with this analysis, since grade 1 cases (most common) were excluded, while per protocol only cases with RX documents were included. This has to result in some bias to more severe cases, as also reflected in the number of cases that underwent surgery (21/50 cases). For this reviewer, and for the future readership it is completely unclear where the 50 cases come from (population admitted, cohort considered, cohort included, inclusion/exclusion criteria, patient flow chart, clinical characteristics and basic information like feeding strategy). This is crucial to ensure absence of bias.
Another relevant method related issue: where those who in retrospect analysed the data blinded for outcome ?
As perforation is a ‘hard’ indication for surgery, how do the results look if cases with co-existing perforation are excluded from the analysis ? and (cf figure 1 of the current version) how were other indicators ‘handled’ like pneumatosis or portal air.
When used as a ‘predictor’, I would like to see some analysis on sensitivity and specificity (and was there consistency between consecutive X rays in a given patient) ? or can this only be used at first assessment ? That the distention is statistically different is ok, but how to use this, any cut off and any data on sensitivity and specificity with such a cut off value.
Minor, editing issues
life-threating (abstract)= please check writing
suggest to refer to the current figure 1 in the method section when describing how the ratio was constructed.
Author Response
I’m somewhat puzzled with this analysis, since grade 1 cases (most common) were excluded, while per protocol only cases with RX documents were included. This has to result in some bias to more severe cases, as also reflected in the number of cases that underwent surgery (21/50 cases). For this reviewer, and for the future readership it is completely unclear where the 50 cases come from (population admitted, cohort considered, cohort included, inclusion/exclusion criteria, patient flow chart, clinical characteristics and basic information like feeding strategy). This is crucial to ensure absence of bias.
ANSWER:
We fully agree with your comments. For clarity reasons, we created a flowchart (figure 1) to show how the patients were selected for the present study. The materials and methods paragraph is also updated accordingly. Inclusion and exclusion criteria are also now added (please see page 3, lines 70-2). We fully agree that this selection of patients may have contributed to the bias towards more severe stages of NEC and leave unanswered the question of the potential importance of intestinal dilatation on initial abdominal radiography in premature NEC stage I patients (please see our reflections on this, page 6, lines 160-1)
Another relevant method related issue: where those who in retrospect analysed the data blinded for outcome ?
ANSWER: Because of the retrospective nature of this study, the data collection and analysis were not blinded for the outcome, which was a potential bias of this study. We briefly discussed it (lines 159-60) and added a paragraph in conclusion (line 164). We also mention other important limitations of our study (lines 158-61). The abstract was also updated accordingly (lines 28-9).
As perforation is a ‘hard’ indication for surgery, how do the results look if cases with co-existing perforation are excluded from the analysis ? and (cf figure 1 of the current version) how were other indicators ‘handled’ like pneumatosis or portal air.
ANSWER: Patients with pneumoperitoneum at the first abdominal radiography were excluded from the study as in these cases the intestinal dilatation was irrelevant radiological data. The patients with radiographic signs of intestinal pneumatosis and portal air were included in the study and further monitored and surgically treated as needed.
When used as a ‘predictor’, I would like to see some analysis on sensitivity and specificity (and was there consistency between consecutive X rays in a given patient) ? or can this only be used at first assessment ? That the distention is statistically different is ok, but how to use this, any cut off and any data on sensitivity and specificity with such a cut off value.
ANSWER: We fully agree with your comments that using the term predictor makes less sense if that statement is not substantiated by determining the sensitivity, specificity and cut off value. Unfortunately, we are not able to present our results as suggested above because we used only the first (initial) radiograph for the measurements presented in the current study. We believe that abdominal radiography with possible detection of intestinal dilatation may be used only in the initial assessment of patients with suspected NEC while consecutive radiographs should be used to monitor potential disease progression as recommended.
Minor, editing issues
life-threating (abstract)= please check writing
ANSWER: Corrected.
suggest to refer to the current figure 1 in the method section when describing how the ratio was constructed.
ANSWER: Corrected.
Round 2
Reviewer 2 Report
only one suggestion
recommend to add the fact that only the first x ray at presentation has been considered in the abstract also, and perhaps even in the title ?
Author Response
Reviewer#2:
Recommend to add the fact that only the first x ray at presentation has been considered in the abstract also, and perhaps even in the title ?
Answer: Thank you. We added "initial" in the title and in the abstract of the manuscript. Again, we wish to thank you for the constructive comments and suggestions that definitely improved the quality of the manuscript.